# Characteristics and outcomes of patients with COVID-19 with and without prevalent hypertension: a multinational cohort study

Carlen Reyes [1], Andrea Pistillo,[1] Sergio Fernández-Bertolín,[1] Martina Recalde,[1,2] Elena Roel,[1,2] Diana Puente,[1,2] Anthony G Sena,[3,4] Clair Blacketer,[3,4] Lana Lai,[5] Thamir M Alshammari,[6] Waheed-UI-Rahman Ahmed,[7,8] Osaid Alser [9], Heba Alghoul [10], Carlos Areia [11], Dalia Dawoud,[12,13] Albert Prats-Uribe [14], Neus Valveny,[15] Gabriel de Maeztu,[16] Luisa Sorlí Redó,[2,17,18] Jordi Martinez Roldan,[19] Inmaculada Lopez Montesinos,[17] Lisa M Schilling,[20] Asieh Golozar,[21,22] Christian Reich,[23] Jose D Posada,[24] Nigam Shah,[24] Seng Chan You,[25] Kristine E Lynch,[26,27] Scott L DuVall,[26,27] Michael E Matheny,[26,27] Fredrik Nyberg,[28] Anna Ostropolets,[29] George Hripcsak,[29,30] Peter R Rijnbeek [4], Marc A Suchard,[31] Patrick Ryan,[3,29] Kristin Kostka [23,32], Talita Duarte-Salles [1]

For numbered affiliations see end of article.

**Correspondence to**
Dr Talita Duarte-Salles;
tduarte@idiapjgol.org

## ABSTRACT

**Objective** To characterise patients with and without prevalent hypertension and COVID-19 and to assess adverse outcomes in both inpatients and outpatients.

**Design and setting** This is a retrospective cohort study using 15 healthcare databases (primary and secondary electronic healthcare records, insurance and national claims data) from the USA, Europe and South Korea, standardised to the Observational Medical Outcomes Partnership common data model. Data were gathered from 1 March to 31 October 2020.

**Participants** Two non-mutually exclusive cohorts were defined: (1) individuals diagnosed with COVID-19 (diagnosed cohort) and (2) individuals hospitalised with COVID-19 (hospitalised cohort), and stratified by hypertension status. Follow-up was from COVID-19 diagnosis/hospitalisation to death, end of the study period or 30 days.

**Outcomes** Demographics, comorbidities and 30-day outcomes (hospitalisation and death for the 'diagnosed' cohort and adverse events and death for the 'hospitalised' cohort) were reported.

**Results** We identified 2 851 035 diagnosed and 563 708 hospitalised patients with COVID-19. Hypertension was more prevalent in the latter (ranging across databases from 17.4% (95% CI 17.2 to 17.6) to 61.4% (95% CI 61.0 to 61.8) and from 25.6% (95% CI 24.6 to 26.6) to 85.9% (95% CI 85.2 to 86.6)). Patients in both cohorts with hypertension were predominantly >50 years old and female. Patients with hypertension were frequently diagnosed with obesity, heart disease, dyslipidaemia and diabetes. Compared with patients without hypertension, patients with hypertension in the COVID-19 diagnosed cohort had more hospitalisations (ranging from 1.3% (95% CI 0.4 to 2.2) to 41.1% (95% CI 39.5 to 42.7) vs from 1.4% (95% CI 0.9 to 1.9) to 15.9% (95% CI 14.9 to 16.9)) and increased mortality (ranging from 0.3% (95% CI 0.1 to 0.5) to 18.5% (95% CI 15.7 to 21.3) vs from 0.2% (95% CI 0.2 to 0.2) to 11.8% (95% CI 10.8 to 12.8)). Patients in the COVID-19 hospitalised cohort with hypertension were more likely to have acute respiratory distress syndrome (ranging from 0.1% (95% CI 0.0 to 0.2) to 65.6% (95% CI 62.5 to 68.7) vs from 0.1% (95% CI 0.0 to 0.2) to 54.7% (95% CI 50.5 to 58.9)), arrhythmia (ranging from 0.5% (95% CI 0.3 to 0.7) to 45.8% (95% CI 42.6 to 49.0) vs from 0.4% (95% CI 0.3 to 0.5) to 36.8% (95% CI 32.7 to 40.9)) and increased mortality (ranging from 1.8% (95% CI 0.4 to 3.2) to 25.1% (95% CI 23.0 to 27.2) vs from 0.7% (95% CI 0.5 to 0.9) to 10.9% (95% CI 10.4 to 11.4)) than patients without hypertension.

## Strengths and limitations of this study

► This study is unique in its approach to characterising COVID-19 cases across an international network of healthcare databases, with diverse healthcare systems and policies, through a comprehensive federated approach.

► This study was carried out using routinely collected clinical practice data, which confer greater external validity, but also imply a risk of misclassification.

► This study was intentionally descriptive and was deliberately not designed for causal inference.

► The diagnosed and/or hospitalised cohorts were non-mutually exclusive.

► The data that underpinned this study mostly came from the initial months of the COVID-19 pandemic and may not be representative of the COVID-19 cases diagnosed and/or hospitalised in the subsequent periods.

**Conclusions** COVID-19 patients with hypertension were more likely to suffer severe outcomes, hospitalisations and deaths compared with those without hypertension.

## INTRODUCTION

As of September 2021, the ongoing COVID-19 pandemic has affected over 220 million people, with an estimated death toll that has surpassed 4.5 million deaths worldwide.[1] Hypertension is a common chronic condition that may increase the risk of hospitalisations and adverse outcomes.[2] A higher prevalence of hypertension has been found among patients with COVID-19 compared with the general population, which has attracted the attention of researchers.[3] The characterisation of this population at risk is key to be able to design effective preventive strategies that could improve patient outcomes and reduce pressure on healthcare systems.

To date, observational studies,[4–16] systematic reviews and meta-analyses have reported an increased risk of progression to severe COVID-19 and increased mortality in patients with hypertension.[17–21] However, these studies either only included hospitalised patients,[4–13 15 16] leading to selection bias, or had a small sample size,[6–10 15] both of which limit extrapolation of results.

Most patients with confirmed SARS-CoV-2 infection experience mild or moderate symptoms (80%)[22] and are predominantly seen as outpatients; therefore, a large characterisation study including both inpatients and outpatients is needed.

This study aims to describe and compare the demographics, baseline comorbidities and 30-day outcomes of individuals with COVID-19 with and without pre-existing hypertension in both inpatients and outpatients.

## MATERIALS AND METHODS
### Study design, setting and data sources

A multinational, multidatabase cohort study was conducted using data from 1 March to 31 October 2020 included in 'The Characterizing Health Associated Risks and Your Baseline Disease In SARS-COV-2' (CHARYBDIS[23]) study. This is a large-scale multinational cohort study aimed to characterise health-associated risks and baseline diseases in SARS-CoV-2 patients using routinely collected primary care and hospital electronic health records (EHR), hospital billing and insurance claims data from the USA, Europe (the Netherlands, Spain, UK, Germany and France) and Asia (South Korea and China).

From the databases contributing to CHARYBDIS, only 20 had available information on pre-existing hypertension and were initially selected. To be included in the study, the databases had to (1) have at least 140 subjects with prevalent hypertension diagnosed with COVID-19 (necessary to estimate the prevalence of previous conditions or 30-day outcomes with sufficient precision; CI width of ±5%) and (2) have at least 1 year of previous

data before the date of COVID-19 diagnosis or hospitalisation. Data results for this paper were extracted on 21 January 2021.[23] Fifteen databases complied with the aforementioned inclusion criteria. Of these, five had data for outpatients: IQVIA-Longitudinal Patients Database 'LPD' (France), IQVIA-Longitudinal Patients Database 'LPD' (Italy), IQVIA-Disease Analyser 'DA' (Germany), Clinical Practice Research Datalink 'CPRD' (UK) and Integrated Primary Care Information 'IPCI' (the Netherlands); two had data for inpatients: Health Insurance Review and Assessment Service 'HIRA' (South Korea) and Hospital del Mar 'HMAR' (Spain); and eight had both inpatient and outpatient data: IQVIA-OpenClaims, HealthVerity, Information System for Research in Primary Care 'SIDIAP' (Spain[24]), Optum de-identified Electronic Health Record Dataset 'OPTUM-HER' (USA), United States Department of Veterans Affairs (VA-OMOP), Columbia University Irving Medical Center 'CUIMC' (USA), University of Colorado Anschutz Medical Campus Health Data Compass 'CU-AMC-HDC', and STAnford Medicine Research Data Repository 'STARR-OMOP' (USA[25]). A more detailed description of the included data sources is available in online supplemental figure 1 and table 1.

### Study participants and follow-up

Two non-mutually exclusive cohorts were defined: (1) individual diagnosed with COVID-19 (COVID-19 diagnosed) and (2) individuals hospitalised with COVID-19 (COVID-19 hospitalised). The COVID-19 diagnosed cohort included individuals with a COVID-19 clinical diagnosis and/or a SARS-CoV-2 positive test. The COVID-19 hospitalised cohort included patients hospitalised with a COVID-19 clinical diagnosis or positive test 21 days before admission up to the end of their hospitalisation. The codes used to identify COVID-19 cases are described in more detail in online supplemental table 2. The index date (ie, cohort start date) was the date of COVID-19 diagnosis or positive test (whichever occurred first) for the diagnosed cohort and the date of hospitalisation for the hospitalised cohort. Cohort participants were followed from the index date to the earliest of death, end of the observation period or 30 days after.

### Baseline characteristics and outcomes of interest

Hypertension diagnosis as well as participants' sex and age were gathered at the index date and identified comorbidities in the year before the index date. Hypertension diagnosis and comorbidities (asthma, cancer, chronic kidney and liver disease, chronic obstructive pulmonary disease, dementia, heart disease, hyperlipidaemia, peripheral vascular disease, type 2 diabetes mellitus, obesity) were ascertained based on the Systematized Nomenclature of Medicine Current Terminology hierarchy, with all descendant codes included. We selected and included comorbidities based on their prevalence in the cohorts of the participating sites

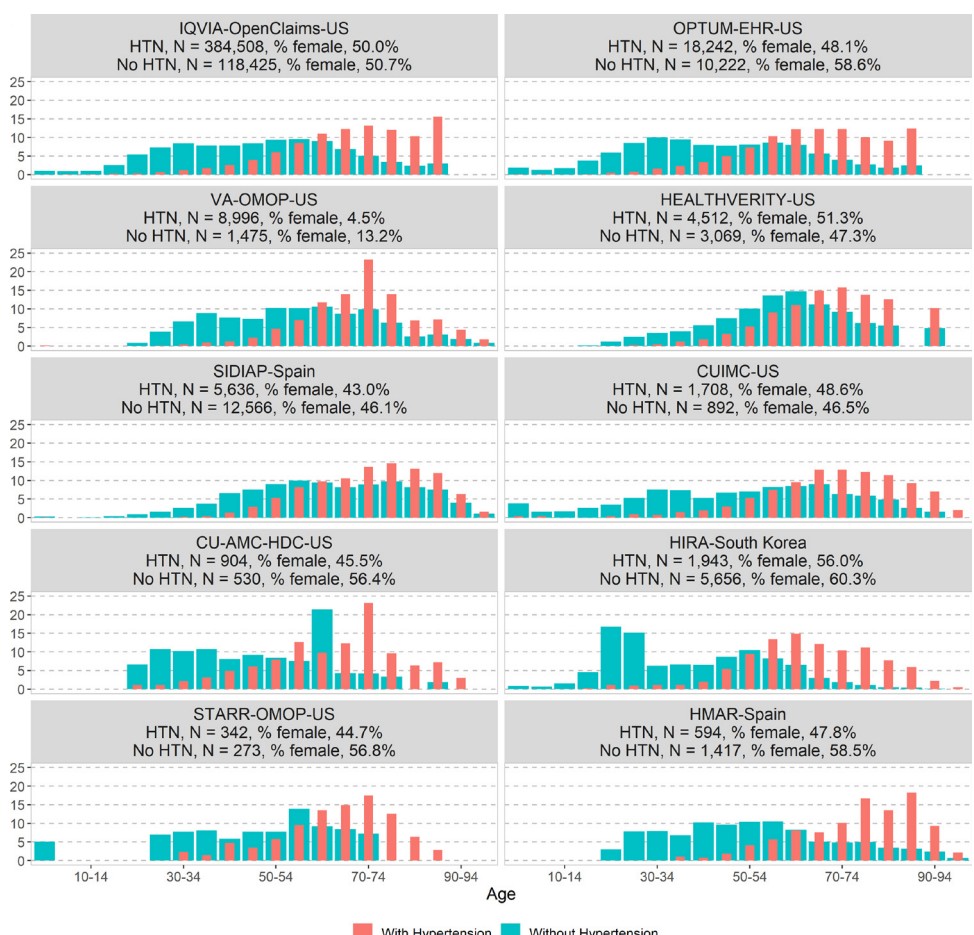

**Figure 1** Comparison of age and sex distribution in patients with a COVID-19 diagnosis with and without hypertension in the CHARYBDIS network, %. CHARYBDIS, Characterizing Health Associated Risks and Your Baseline Disease In SARS-COV-2; CUIMC, Columbia University Irving Medical Center; HIRA, Health Insurance Review and Assessment Service; HMAR, Hospital del Mar; HTN, hypertension; OMOP, Observational Medical Outcomes Partnership; OPTUM-EHR, Optum de-identified Electronic Health Record Dataset; SIDIAP, Information System for Research in Primary Care; STARR-OMOP, STAnford Medicine Research Data Repository; VA-OMOP, United States Department of Veterans Affairs; CU-AMC-HDC, University of Colorado Anschutz Medical Campus Health Data Compass

and their clinical relevance to the COVID-19 research field.[17–21] Clinical epidemiologists generated a list of codes for the identification of prior medical conditions and outcomes of interest using a web-based integrated platform (ATLAS tool; https://atlas.ohdsi.org/). The definition of the variables can be found in online supplemental table 3.

Our main 30-day outcomes of interest were hospitalisation and death for the COVID-19 diagnosed cohort, and requirement for intensive services (identified as any record of mechanical ventilation and/or tracheostomy and/or extracorporeal membrane oxygenation procedure), acute respiratory distress syndrome (ARDS), arrhythmia, total cardiovascular events (ischaemic stroke, haemorrhagic stroke, heart failure (heart failure during hospitalisation for the hospitalised cohort), acute myocardial infarction or sudden cardiac death), sepsis, venous thromboembolism and death for the COVID-19 hospitalised cohort.

## Statistical analyses

All data were standardised to the Observational Medical Outcomes Partnership common data model.[26] A common analytical code for the CHARYBDIS study was developed for the Observational Health Data Sciences and Informatics Methods Library, which was run locally in each database. Only aggregate results from each database were publicly shared. The CHARYBDIS protocol and source code can be found at https://githubcom/ohdsi-studies/Covid19CharacterizationCharybdis.

Demographics (sex and age categorised in 5-year age bands), comorbidities and 30-day incidence rates of outcomes were reported as proportion along with 95% CI. A minimum of five individuals was established to minimise the risk of identification of patients.

All results are reported by cohort, database and hypertension status (with or without hypertension).

This is a descriptive study and no causal inference is intended. Multivariable regression or adjustment for confounding was therefore considered out of remit and

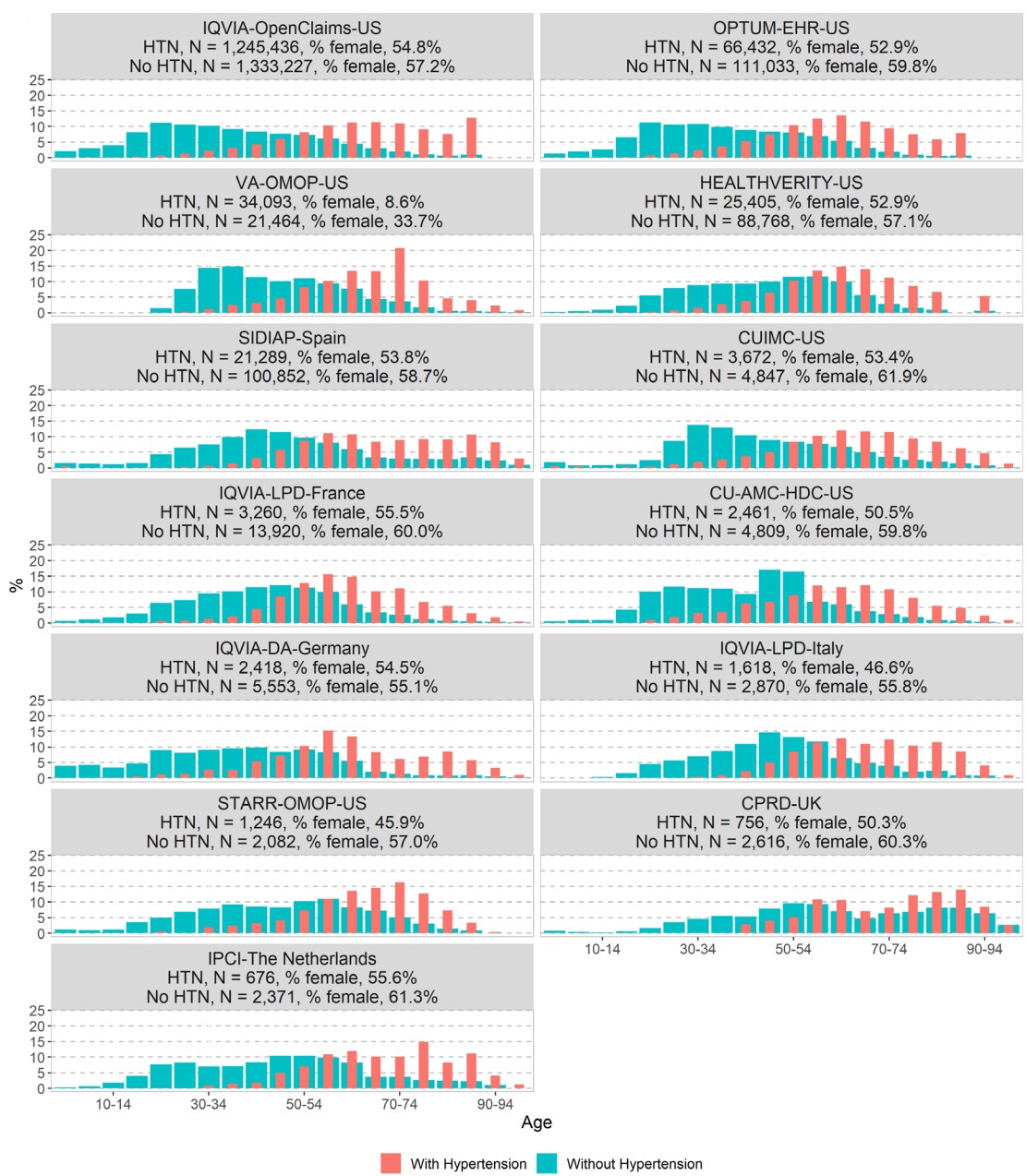

**Figure 2** Comparison of age and sex distribution in patients with a COVID-19 hospitalisation with and without hypertension in the CHARYBDIS network, %. CHARYBDIS, Characterizing Health Associated Risks and Your Baseline Disease In SARS-COV-2; CPRD, Clinical Practice Research Datalink; CUIMC, Columbia University Irving Medical Center; DA, Disease Analyser; HTN, hypertension; IPCI, Integrated Primary Care Information; LPD, Longitudinal Patients Database; OMOP, Observational Medical Outcomes Partnership; OPTUM-EHR, Optum de-identified Electronic Health Record Dataset; SIDIAP, Information System for Research in Primary Care; STARR-OMOP, STAnford Medicine Research Data Repository; VA-OMOP, United States Department of Veterans Affairs; CU-AMC-HDC, University of Colorado Anschutz Medical Campus Health Data Compass.

not included in our study. We used R V.4.0.3 for data visualisation. All data partners consented to the external sharing of the result set on data.ohdsi.org. Consent to participate was not required as only anonymised retrospective data were used for this study and no patient or general practitioner contact was required.

**Patient and public involvement**

No patients were involved.

**RESULTS**

**Study population**

Overall, 2 851 035 patients diagnosed and 563 708 patients hospitalised with COVID-19 were identified in 15 databases from 8 countries (USA, South Korea, Germany, the Netherlands, France, Italy, Spain and UK). In total, 1 408 762 and 427 385 patients diagnosed and hospitalised with COVID-19, respectively, had a prior diagnosis of hypertension (online supplemental

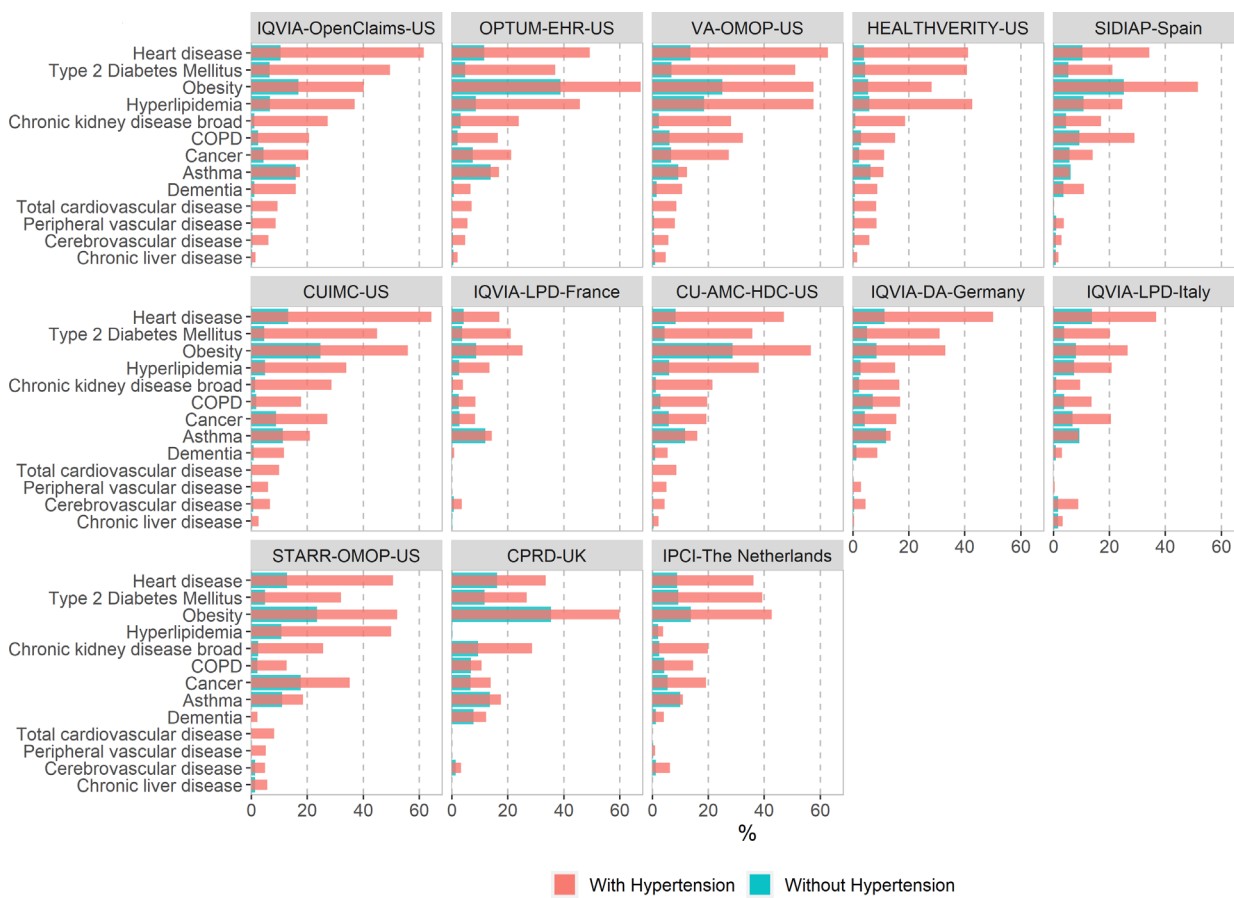

**Figure 3** Comorbidities at baseline among patients with a COVID-19 diagnosis with and without hypertension in the CHARYBDIS network, %. CHARYBDIS, Characterizing Health Associated Risks and Your Baseline Disease In SARS-COV-2; COPD, chronic obstructive pulmonary disease; CPRD, Clinical Practice Research Datalink; CUIMC, Columbia University Irving Medical Center ; DA, Disease Analyser; IPCI, Integrated Primary Care Information; LPD, Longitudinal Patients Database; OMOP, Observational Medical Outcomes Partnership; OPTUM-EHR, Optum de-identified Electronic Health Record Dataset; SIDIAP, Information System for Research in Primary Care; STARR-OMOP, STAnford Medicine Research Data Repository; VA-OMOP, United States Department of Veterans Affairs; CU-AMC-HDC, University of Colorado Anschutz Medical Campus Health Data Compass.

table 4). The prevalence of hypertension ranged from 17.4% to 48.3% in the COVID-19 diagnosed cohort and from 25.6% to 85.9% in the COVID-19 hospitalised cohort.

### Baseline characteristics

The age and sex distributions in the COVID-19 diagnosed cohort and in the COVID-19 hospitalised cohort with and without hypertension are presented in figures 1 and 2, respectively. Overall, in both cohorts, patients with hypertension were older than those without (higher proportion of patients aged above 50 across all databases). The proportion of patients diagnosed with COVID-19 and hypertension peaked at a younger age (55–70 years old) compared with those hospitalised (70–80 years old). The proportion of women with hypertension was greater in the diagnosed cohort (8.6%–55.6%) than in the hospitalised cohort (4.5%–56%).

### Baseline comorbidities

Figures 3 and 4 report the proportion of baseline comorbidities of the COVID-19 diagnosed cohort (figure 3) and the COVID-19 hospitalised cohort (figure 4) with and without hypertension. Patients with hypertension and COVID-19 diagnosed or hospitalised were frequently diagnosed with obesity, heart disease, dyslipidaemia and type 2 diabetes, the proportion of which more than doubles the ones found among patients with COVID-19 without hypertension.

### 30-day outcomes of interest

The 30-day outcomes in people with and without hypertension in both the COVID-19 diagnosed and/or hospitalised cohorts are reported in tables 1 and 2.

Patients with hypertension diagnosed with COVID-19 were more likely to be hospitalised (range 1.3%–41.1% vs 1.4%–15.9%) and had increased mortality (range 0.3%–18.5% vs 0.2%–11.8%) when compared with those without hypertension (table 1).

Patients with hypertension hospitalised with COVID-19 were more frequently diagnosed with ARDS (range 0.1%–65.6% vs 0.1%–54.7%) and cardiac arrhythmia (range 0.5%–45.8% vs 0.4%–36.8%) and had increased mortality (range 1.8%–25.1% vs 0.7%–10.9%) as compared with those without hypertension (table 2).

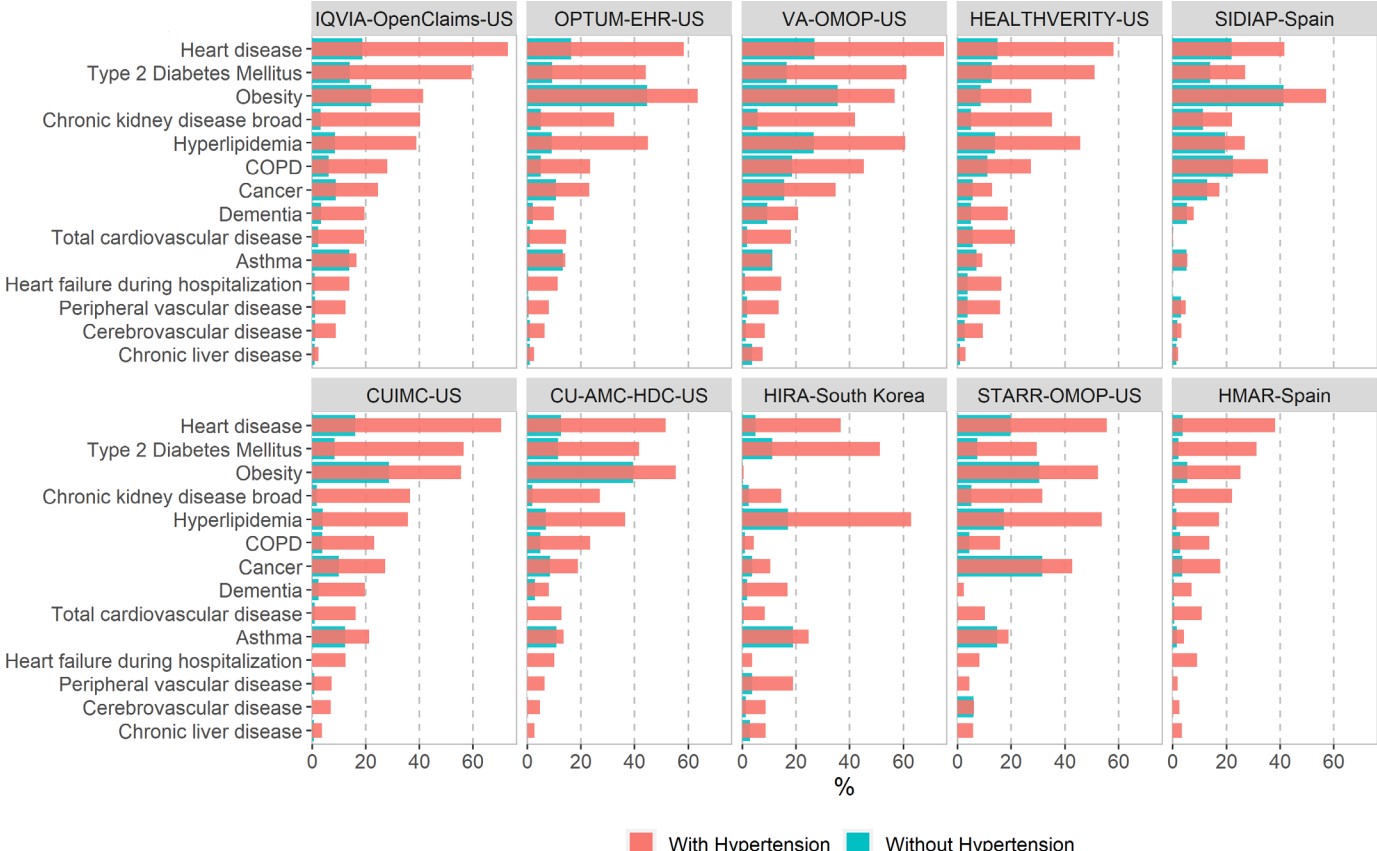

**Figure 4** Comorbidities at baseline among patients with a COVID-19 hospitalisation with and without hypertension in the CHARYBDIS network, %. CHARYBDIS, Characterizing Health Associated Risks and Your Baseline Disease In SARS-COV-2; COPD, chronic obstructive pulmonary disease; CUIMC, Columbia University Irving Medical Center; HIRA, Health Insurance Review and Assessment Service; HMAR, Hospital del Mar; OMOP, Observational Medical Outcomes Partnership; OPTUM-EHR, Optum de-identified Electronic Health Record Dataset; SIDIAP, Information System for Research in Primary Care; STARR-OMOP, STAnford Medicine Research Data Repository; VA-OMOP, United States Department of Veterans Affairs; CU-AMC-HDC, University of Colorado Anschutz Medical Campus Health Data Compass.

## DISCUSSION

This large multinational, multidatabase cohort study reports a greater prevalence of hypertension among patients hospitalised with COVID-19 compared with those diagnosed with COVID-19. Patients with hypertension diagnosed and/or hospitalised with COVID-19 were frequently diagnosed with obesity, heart disease, dyslipidaemia and type 2 diabetes at baseline compared with those without hypertension. They were also more likely to experience adverse outcomes including death and hospitalisations (in the COVID-19 diagnosed cohort) and cardiac arrhythmia, ARDS and death (in the COVID-19 hospitalised cohort) than patients without hypertension.

This is the first large multinational study that characterises both inpatients and outpatients with COVID-19 with and without prevalent hypertension. Hypertension was more prevalent in hospitalised patients compared with those diagnosed with COVID-19 (range 25.6%–85.9% vs 17.4%–61.4%, respectively). The observed variability between databases is similar to previous reports, where the prevalence ranged from 28.8%[7] to 60%.[15]

However, these results should be put into context given that our highest rate (in both COVID-19 diagnosed and COVID-19 hospitalised) was observed in the VA-OMOP database from the US Department of Veterans Affairs (mostly men of older age).

As in the general population with hypertension,[27] patients with hypertension diagnosed with COVID-19 in this study were more frequently diagnosed with heart disease or type 2 diabetes at baseline than individuals without hypertension. These results are similar to what has been previously published, where patients with hypertension and COVID-19 also reported a higher prevalence of diabetes mellitus,[6 8 12] cardiovascular diseases (other than hypertension)[8 12] and chronic kidney disease[8] compared with those without hypertension. This study further expands these previous findings identifying these same comorbidities in the outpatients diagnosed with COVID-19 and adds obesity and dyslipidaemia to the list of conditions more frequently found among patients with COVID-19 and hypertension compared with those without hypertension. The higher prevalence of comorbid conditions found in this study among patients with hypertension hospitalised with COVID-19 compared with patients with hypertension diagnosed with COVID-19 suggests a poorer baseline health status.

**Table 1** Comparison of 30-day outcomes of interest between patients with COVID-19 with and without hypertension in the COVID-19 diagnosed cohorts in the CHARYBDIS network

| Database | Hypertension | n | 30-day outcomes, % (95% CI) | |
| --- | --- | --- | --- | --- |
| | | | Death | Hospitalisation |
| IQVIA-OpenClaims (USA) | With | 1 245 436 | – | 29.6 (29.5 to 29.7) |
| | Without | 1 333 227 | – | 8.9 (8.9 to 8.9) |
| OPTUM-HER (USA) | With | 66 432 | 1.7 (1.6 to 1.8) | 26.4 (26.1 to 26.7) |
| | Without | 111 033 | 0.2 (0.2 to 0.2) | 9.2 (9.0 to 9.4) |
| VA-OMOP (USA) | With | 34 093 | 5.4 (5.2 to 5.6) | 23.4 (23.0 to 23.8) |
| | Without | 21 464 | 0.7 (0.6 to 0.8) | 6.1 (5.8 to 6.4) |
| HealthVerity (USA) | With | 25 405 | – | 14.6 (14.2 to 15.0) |
| | Without | 88 768 | – | 3.1 (3.0 to 3.2) |
| SIDIAP (Spain) | With | 21 289 | 9.8 (9.4 to 10.2) | 22.8 (22.2 to 23.4) |
| | Without | 100 852 | 3.3 (3.2 to 3.4) | 11.2 (11.0 to 11.4) |
| CUIMC (USA) | With | 3672 | 11.8 (10.8 to 12.8) | 41.1 (39.5 to 42.7) |
| | Without | 4847 | 2.0 (1.6 to 2.4) | 15.9 (14.9 to 16.9) |
| CU-AMC-HDC (USA) | With | 2461 | 5.9 (5.0 to 6.8) | 35.8 (33.9 to 37.7) |
| | Without | 4809 | 0.7 (0.5 to 0.9) | 11.2 (10.3 to 12.1) |
| IQVIA-DA (Germany) | With | 2418 | 0.3 (0.1 to 0.5) | – |
| | Without | 5553 | – | – |
| STARR-OMOP (USA) | With | 1246 | 0.6 (0.2 to 1.0) | 24.6 (22.2 to 27.0) |
| | Without | 2082 | – | 14.0 (12.5 to 15.5) |
| CPRD (UK) | With | 756 | 18.5 (15.7 to 21.3) | – |
| | Without | 2616 | 11.8 (10.6 to 13.0) | – |
| IPCI (The Netherlands) | With | 676 | 13.6 (11.0 to 16.2) | 1.3 (0.4 to 2.2) |
| | Without | 2371 | 3.1 (2.4 to 3.8) | 1.4 (0.9 to 1.9) |

'–' means information is not available or <5 cases for all databases except for CU-AMC HDC, where information is not available for <10 cases.
CHARYBDIS, Characterizing Health Associated Risks and Your Baseline Disease In SARS-COV-2; CPRD, Clinical Practice Research Datalink; CU-AMC-HDC, University of Colorado Anschutz Medical Campus Health Data Compass; CUIMC, Columbia University Irving Medical Center; DA, Disease Analyser; IPCI, Integrated Primary Care Information; OMOP, Observational Medical Outcomes Partnership; OPTUM-HER, Optum de-identified Electronic Health Record Dataset; SIDIAP, Information System for Research in Primary Care; STARR, STAnford Medicine Research Data Repository; VA-OMOP, United States Department of Veterans Affairs.

Patients with hypertension hospitalised with COVID-19 were more likely to have worse disease progression with higher rates of ARDS (prevalence per cent change (PC) between patients with and without hypertension ranging from –1.8% to 18.2%), more cardiac arrhythmias (PC ranging from 0.1% to 20.5%) and increased mortality (PC ranging from 3.5% to 14.9%). Previous studies have documented poorer clinical outcomes in patients with hypertension hospitalised with COVID-19 (including ARDS),[8 12 14 19] the need for mechanical ventilation, admission to intensive care units[6 13 19] or an increased mortality.[4 7 11–13 19] This study further showed that patients with hypertension diagnosed with COVID-19 were more likely to experience hospitalisations (PC between patients with and without hypertension ranging from –0.1% to 25.6%) and deaths (PC from 1.5% to 10.5%).

These results highlight the importance of considering hypertension as a possible risk factor in the overall population diagnosed and not only in those hospitalised with COVID-19. It also adds to the current literature cardiac arrhythmia and cardiovascular diseases (other than hypertension) to the list of adverse outcomes more frequently diagnosed among patients with hypertension hospitalised with COVID-19 compared with those without hypertension.

This study has several strengths. This is the largest cohort study on individuals with hypertension who were diagnosed and/or hospitalised with COVID-19 to date. It provides novel insight into the characterisation of patients diagnosed with COVID-19 and confers greater external validity of its results compared with what has been published up to date (only hospitalised patients). It is also unique in its approach to characterising COVID-19 cases across an international network of healthcare databases, with diverse healthcare systems and policies, through a comprehensive federated approach, allowing analysis of 15 databases without sharing patient identifiable data, hence respecting patients' confidentiality at all times.

We recognise there are limitations to our approach. First, this study was intentionally descriptive and was deliberately not designed for causal inference. The

**Table 2** Comparison of 30-day outcomes of interest between patients with COVID-19 with and without hypertension in the COVID-19 hospitalised cohorts in the CHARYBDIS network

| Database | Hypertension | n | 30-day outcomes, % (95% CI) | | | | | | |
|---|---|---|---|---|---|---|---|---|---|
| | | | VTE* | Death | Cardiac arrhythmia | Sepsis | ARDS | Intensive services | Total CVE† |
| IQVIA-OpenClaims (USA) | With | 384508 | 3.9 (3.8 to 4.0) | – | 15.4 (15.3 to 15.5) | 18.3 (18.2 to 18.4) | 34.8 (34.6 to 35.0) | 9.1 (9.0 to 9.2) | 11.3 (11.2 to 11.4) |
| | Without | 118425 | 3.8 (3.7 to 3.9) | – | 7.2 (7.1 to 7.3) | 15.5 (15.3 to 15.7) | 31.3 (31.0 to 31.6) | 6.4 (6.3 to 6.5) | 4.5 (4.4 to 4.6) |
| OPTUM-HER (USA) | With | 18242 | 6.2 (5.9 to 6.5) | 5.1 (4.8 to 5.4) | 31.6 (30.9 to 32.3) | 24.8 (24.2 to 25.4) | 45.7 (45.0 to 46.4) | 14.0 (13.5 to 14.5) | 18.2 (17.6 to 18.8) |
| | Without | 10222 | 4.4 (4.0 to 4.8) | 1.6 (1.4 to 1.8) | 11.1 (10.5 to 11.7) | 15.0 (14.3 to 15.7) | 27.5 (26.6 to 28.4) | 6.3 (5.8 to 6.8) | 4.8 (4.4 to 5.2) |
| VA-OMOP (USA) | With | 8996 | 7.3 (6.8 to 7.8) | 15.4 (14.7 to 16.1) | 33.9 (32.9 to 34.9) | 20.0 (19.2 to 20.8) | 43.9 (42.9 to 44.9) | 17.1 (16.3 to 17.9) | 21.0 (20.2 to 21.8) |
| | Without | 1475 | 6.9 (5.6 to 8.2) | 7.6 (6.2 to 9.0) | 16.8 (14.9 to 18.7) | 16.2 (14.3 to 18.1) | 39.6 (37.1 to 42.1) | 11.2 (9.6 to 12.8) | 7.3 (6.0 to 8.6) |
| HealthVerity (USA) | With | 4512 | 3.6 (3.1 to 4.1) | – | 14.8 (13.8 to 15.8) | 16.5 (15.4 to 17.6) | 26.7 (25.4 to 28.0) | 6.1 (5.4 to 6.8) | 11.9 (11.0 to 12.8) |
| | Without | 3069 | 3.9 (3.2 to 4.6) | – | 6.8 (5.9 to 7.7) | 12.5 (11.3 to 13.7) | 23.9 (22.4 to 25.4) | 4.9 (4.1 to 5.7) | 5.6 (4.8 to 6.4) |
| SIDIAP (Spain) | With | 5636 | 1.0 (0.7 to 1.3) | 15.4 (14.5 to 16.3) | 0.5 (0.3 to 0.7) | – | 0.1 (0.0 to 0.2) | – | 0.9 (0.7 to 1.1) |
| | Without | 12566 | 1.1 (0.9 to 1.3) | 10.9 (10.4 to 11.4) | 0.4 (0.3 to 0.5) | 0.0 (0.0 to 0.0) | 0.1 (0.0 to 0.2) | – | 0.5 (0.4 to 0.6) |
| CUIMC (USA) | With | 1708 | 3.9 (3.0 to 4.8) | 25.1 (23.0 to 27.2) | 12.1 (10.6 to 13.6) | 6.1 (5.0 to 7.2) | 16.0 (14.3 to 17.7) | 2.2 (1.5 to 2.9) | 8.1 (6.8 to 9.4) |
| | Without | 892 | 3.6 (2.4 to 4.8) | 10.2 (8.2 to 12.2) | 4.7 (3.3 to 6.1) | 5.3 (3.8 to 6.8) | 17.8 (15.3 to 20.3) | 1.8 (0.9 to 2.7) | 3.8 (2.5 to 5.1) |
| CU-AMC HDC (USA) | With | 904 | 11.4 (9.3 to 13.5) | 14.9 (12.6 to 17.2) | 45.8 (42.6 to 49.0) | 34.2 (31.1 to 37.3) | 65.6 (62.5 to 68.7) | 28.3 (25.4 to 31.2) | 19.8 (17.2 to 22.4) |
| | Without | 530 | 6.0 (4.0 to 8.0) | 6.0 (4.0 to 8.0) | 36.8 (32.7 to 40.9) | 27.4 (23.6 to 31.2) | 54.7 (50.5 to 58.9) | 15.5 (12.4 to 18.6) | 5.7 (3.7 to 7.7) |
| HIRA (South Korea) | With | 1943 | 0.7 (0.3 to 1.1) | 7.7 (6.5 to 8.9) | 4.4 (3.5 to 5.3) | 5.3 (4.3 to 6.3) | 2.6 (1.9 to 3.3) | 4.9 (3.9 to 5.9) | 10.0 (8.7 to 11.3) |
| | Without | 5656 | NC | 0.7 (0.5 to 0.9) | 0.7 (0.5 to 0.9) | 3.1 (2.6 to 3.6) | 0.5 (0.3 to 0.7) | 0.6 (0.4 to 0.8) | 4.7 (4.1 to 5.3) |
| STARR-OMOP (USA) | With | 342 | 2.0 (0.5 to 3.5) | 1.8 (0.4 to 3.2) | 22.2 (17.8 to 26.6) | 9.9 (6.7 to 13.1) | 12.6 (9.1 to 16.1) | 9.1 (6.1 to 12.1) | 16.4 (12.5 to 20.3) |
| | Without | 273 | NC | – | 6.6 (3.7 to 9.5) | 7.0 (4.0 to 10.0) | 11.4 (7.6 to 15.2) | 5.5 (2.8 to 8.2) | – |
| HMAR (Spain) | With | 594 | 3.2 (1.8 to 4.6) | 14.3 (11.5 to 17.1) | 23.1 (19.7 to 26.5) | 1.9 (0.8 to 3.0) | 12.6 (9.9 to 15.3) | 13.5 (10.8 to 16.2) | 12.1 (9.5 to 14.7) |
| | Without | 1417 | 2.6 (1.8 to 3.4) | 3.9 (2.9 to 4.9) | 6.6 (5.3 to 7.9) | 0.7 (0.3 to 1.1) | 7.3 (5.9 to 8.7) | 6.6 (5.3 to 7.9) | 2.2 (1.4 to 3.0) |

'–' means information is not available or <5 cases for all databases except for CU-AMC HDC, where information is not available for <10 cases.

*Venous thromboembolic (pulmonary embolism and deep vein thrombosis) events.

†Cardiovascular disease events (ischaemic stroke, haemorrhagic stroke, heart failure (heart failure during hospitalisation for the hospitalised cohort), acute myocardial infarction or sudden cardiac death).

ARDS, acute respiratory distress syndrome; CHARYBDIS, Characterizing Health Associated Risks and Your Baseline Disease In SARS-COV-2; CU-AMC-HDC, University of Colorado Anschutz Medical Campus Health Data Compass; CUIMC, Columbia University Irving Medical Center; HIRA, Health Insurance Review and Assessment Service; HMAR, Hospital del Mar; OMOP, Observational Medical Outcomes Partnership; OPTUM-HER, Optum de-identified Electronic Health Record Dataset; SIDIAP, Information System for Research in Primary Care; STARR, STAnford Medicine Research Data Repository; VA-OMOP, United States Department of Veterans Affairs.

observed differences between groups (eg, with vs without hypertension) should therefore not be interpreted as causal effects. Our patients were analysed depending on whether they were diagnosed and/or hospitalised with COVID-19 according to database registration procedures; however, variations could have occurred during the processes by which patients were screened, tested, admitted and registered across time and the databases. Additionally, the diagnosed and/or hospitalised cohorts were non-mutually exclusive and therefore could be patients in the diagnosed cohort who were also hospitalised and vice versa.

This study was carried out using data recorded in routine clinical practice based on EHRs and/or claims; therefore, data could be incomplete or be erroneous, leading to potential misclassification. We have therefore selectively reported database-specific outcomes to minimise the impact of incompleteness. Differential reporting in databases is likely due to different coding practices, different primary-level and secondary-level data availability, as well as variability in disease severity, with milder/less symptomatic cases more likely being only diagnosed and more severe ones hospitalised. Finally, the data that underpinned this study mostly came from the initial months of the COVID-19 pandemic and may not be representative of the COVID-19 cases diagnosed and/or hospitalised in the subsequent periods.

## CONCLUSIONS

Patients with COVID-19 diagnosed with hypertension are more likely to have comorbidities, experience more severe outcomes including hospitalisations and deaths (among outpatients with COVID-19), and experience more ARDS and deaths (among inpatients with COVID-19) compared with patients without hypertension.

**Author affiliations**
[1]Fundació Institut Universitari per a la recerca a l'Atenció Primària de Salut Jordi Gol i Gurina (IDIAPJGol), Barcelona, Spain
[2]Universitat Autonoma de Barcelona, Barcelona, Spain
[3]Janssen Research and Development Titusville, Titusville, New Jersey, USA
[4]Department of Medical Informatics, Erasmus University Medical Center, Rotterdam, The Netherlands
[5]School of Medical Sciences, The University of Manchester, Manchester, UK
[6]College of Pharmacy, Riyadh Elm University, Riyadh, Saudi Arabia
[7]Nuffield Department of Orthopaedics, Rheumatology and Musculoskeletal Sciences, University of Oxford, Botnar Research Center, Oxford, UK
[8]College of Medicine and Health, University of Exeter, St Luke's Campus, Exeter, UK
[9]Massachusetts General Hospital, Harvard Medical School, Boston, Massachusetts, USA
[10]Faculty of Medicine, Islamic University of Gaza, Gaza, Palestine
[11]Nuffield Department of Clinical Neurosciences, University of Oxford, Oxford, UK
[12]National Institute for Health and Care Excellence (NICE), London, UK
[13]Faculty of Pharmacy, Cairo University, Cairo, Egypt
[14]Center for Statistics in Medicine, NDORMS, University of Oxford, Botnar Research Center, Nuffield Orthopaedic Center, Oxford, UK
[15]Real-World Evidence, TFS, Barcelona, Spain
[16]IOMED, Barcelona, Spain
[17]Department of Infectious Diseases, Hospital del Mar, Institut Hospital del Mar d'Investigació Mèdica (IMIM), Barcelona, Spain
[18]Universitat Pompeu Fabra, Barcelona, Spain
[19]Director of Innovation and Digital Transformation, Hospital del Mar, Barcelona, Spain
[20]University of Colorado - Anschutz Medical Campus, Aurora, Colorado, USA
[21]Johns Hopkins University Bloomberg School of Public Health, Baltimore, Maryland, USA
[22]Regeneron Pharmaceuticals, Tarrytown, NY, USA
[23]Real-World Solutions, IQVIA, Cambridge, Massachusetts, USA
[24]Stanford University School of Medicine, Stanford, California, USA
[25]Department of Preventive Medicine, Yonsei University College of Medicine, Seoul, Korea (the Republic of)
[26]VA Informatics and Computing Infrastructure, VA Salt Lake City Health Care System, Salt Lake City, Utah, USA
[27]Department of Internal Medicine, The University of Utah School of Medicine, Salt Lake City, Utah, USA
[28]School of Public Health and Community Medicine, Institute of Medicine, Sahlgrenska Academy, University of Gothenburg, Gothenburg, Sweden
[29]Department of Biomedical Informatics, Columbia University Irving Medical Center, New York, New York, USA
[30]Medical Informatics Services, New York-Presbyterial Hospital, New York, NY, USA
[31]Department of Biostatistics, Fielding School of Publich Health, University of California, Los Angeles, California, USA
[32]The OHDSI Center at the Roux Institute, Northeastern University, Portland, ME, USA

**Acknowledgements** We would like to acknowledge the patients who suffered from or died of this devastating disease and their families and carers. We also thank the healthcare professionals involved in the management of COVID-19 during these challenging times, from primary care to intensive care units. We appreciate the Korean Health Insurance Review and Assessment Service for providing data. We also thank the database curation teams around the world, including the COVIDMAR Group (JP Horcajada, R Güerri, J Villar, M Montero, S Gómez-Zorrilla, M Arenas-Miras, J Gómez-Junyent, I Arrieta, E Sendra, S Castañeda, E Letang, I Pelegrín, A Rial, J Rodríguez, C Gimenez, J Soldado, E García). We also thank the important contribution to this work of Dr Daniel Prieto-Alhambra.

**Contributors** CRey, AGS, CA, AP-U, AG, FN, AO, GH, PRR, KK, TD-S, KEL, SLD, MR, ER, SF-B and AP provided substantial contribution to the conception or design, analysis and interpretation of data for the work. CRey, AGS, CA, AP-U, AG, FN, AO, GH, PRR, KK, TD-S, KEL, SLD, MR, ER, SF-B, AP, DP, NV, GdM, LSR, JMR, ILM, NS, PR, MAS, MEM, CB, LL, TMA, W-U-RA, OA, HA and DD drafted or revised the manuscript critically for important intellectual content. All authors approved the final version of the manuscript. CRey, AGS, CA, AP-U, AG, FN, AO, GH, PRR, KK, TD-S, LL, TMA, W-U-RA, OA, HA, DD, LMS, CRei, JDP and SCY agreed to be accountable for all aspects of the work (KEL and SLD only for VA data) in ensuring that questions related to the accuracy or integrity of any part of the work are appropriately investigated and resolved. TD-S acted as the guarantor and accepted full responsibility for the finished work, had access to the data, and controlled the decision to publish.

**Funding** This work was supported by several funders as follows: the European Health Data and Evidence Network received funding from the Innovative Medicines Initiative 2 Joint Undertaking (JU) under grant agreement number 806968. The JU received support from the European Union's Horizon 2020 research and innovation programme and EFPIA. This research received partial support from the National Institute for Health Research (NIHR) Oxford Biomedical Research Centre (BRC), US National Institutes of Health (R01 LM006910), US Department of Veterans Affairs, the Health Department from the Generalitat de Catalunya with a grant for research projects on SARS-CoV-2 and COVID-19 disease organised by the Direcció General de Recerca i Innovació en Salut, Janssen Research and Development, TFS, IOMED and IQVIA. The University of Oxford received funding related to this work from the Bill and Melinda Gates Foundation (Investment ID INV-016201 and INV-019257). TFS received funding related to this work from the University of Oxford. This work was also supported with funding (resources and facilities) of the Department of Veterans Affairs (VA) Informatics and Computing Infrastructure (VINCI) (VA HSR RES 13-457).

**Disclaimer** No funders had a direct role in this study. The views and opinions expressed are those of the authors and do not necessarily reflect those of the Clinician Scientist Award programme, NIHR, Department of Veterans Affairs or the US Government, NHS, or the Department of Health, England.

**Competing interests** SLD reports grants from Anolinx. MAS reports grants from US National Institutes of Health, grants from Department of Veterans Affairs, during

the conduct of the study; grants from IQVIA, personal fees from Janssen Research and Development, grants from US Food and Drug Administration, personal fees from Private Health Management, outside the submitted work. GH reports grants from NIH, during the conduct of the study; and grants from Janssen Research, outside the submitted work. FN reports being an employee of AstraZeneca until 2019 and holds some AstraZeneca shares, outside the submitted work. KK reports personal fees from the National Institutes of Health, outside the submitted work, and at the time of data analysis and initial drafting of the manuscript. KK was an employee of IQVIA. CRei reports he is an employee of IQVIA. GdM is an employee of IOMED. NV is an employee of TFS. AGS reports personal fees from Janssen R&D, outside the submitted work, and is a full-time employee of Janssen R&D and a Johnson and Johnson shareholder. CB reports personal fees from Janssen R&D, outside the submitted work, and is a full-time employee of Janssen R&D and a Johnson and Johnson shareholder. JDP reports grants from the National Library of Medicine, during the conduct of the study. AG is an employee of Regeneron Pharmaceuticals and reports stocks from Regeneron Pharmaceuticals. PRR reports having received research group grants from Innovative Medicine Initiative and Janssen Research and Development. PR reports being an employee of Janssen Research and Development and a shareholder of Johnson & Johnson. ER, SF-B, NS, LMS, DP, SCY, MR, AP-U, HA, KEL, MEM, AO, CA, CRey, TD-S, TMA, OA, W-U-RA, ILM, JMR, LSR, DD, LL and AP have nothing to declare. No other relationships or activities could appear to have influenced the submitted work.

**Patient consent for publication** Not required.

**Ethics approval** Before performing these analyses, all the data partners received Institutional Review Board (IRB) approval or exemption. STARR-OMOP had approval from IRB Panel #8 (RB-53248) registered to Leland Stanford Junior University under the Stanford Human Research Protection Program (HRPP). The use of VA-OMOP data was reviewed by the Department of Veterans Affairs Central IRB, was determined to meet the criteria for exemption under Exemption Category 4(3), and approved for waiver of HIPAA Authorization. The use of SIDIAP was approved by the Clinical Research Ethics Committee of the IDIAPJGol (project code: 20/070-PCV). The use of CPRD was approved by the Independent Scientific Advisory Committee (ISAC) (protocol number 20_059RA2). The use of the CUIMC database was approved by the Columbia University Institutional Review Board as an OHDSI network study (IRB-AAAO7805). The use of HMAR was approved by the Parc de Salut Mar Clinical Research Ethics Committee (Comité de Ética de la Investigación con medicamentos del Parc de Salut MAR, IRB-2020/9183). The use of HIRA database was approved by the IRB of Ajou University ('AJIRB-MED-EXP-20-061'). The Colorado Multiple Institutional Review Board, CB F490 University of Colorado, Anschutz Medical Campus extended an exemption of IRB certificate on 17 November 2020 for the use of the CU-AMC-HDC data for this study. Moreover, given that this study only used de-identified data with no transmission of patient-level information at any time during the analysis and that all data reported were aggregated and no identification of individual patients or physicians was possible, some databases (IQVIA-OpenClaims, IQVIA DA Germany, IQVIA LPD France, IQVIA LPD Italy, IPCI) considered this study as not human subject research and therefore, no further approval was necessary. Furthermore, the New England Institutional Review Board of Janssen Research and Development (Raritan, New Jersey) has determined that studies conducted on licensed copies of Optum EHR and HealthVerity are exempt from study-specific IRB review as these studies do not qualify as human subjects' research.

**Provenance and peer review** Not commissioned; externally peer reviewed.

**Data availability statement** Data may be obtained from a third party and are not publicly available. Open Science is a guiding principle within OHDSI. As such, we provide unfettered access to all open-source analysis tools employed in this study via https://github.com/ohdsi-studies/Covid19CharacterizationCharybdis, as well as all data and results artefacts that do not include patient-level health information via https://data.odhsi.org/Covid19CharacterizationCharybdis/. Data partners contributing to this study remain custodians of their individual patient-level health information and hold either IRB exemption or approval for participation.

**ORCID iDs**
Carlen Reyes http://orcid.org/0000-0001-8486-3265
Osaid Alser http://orcid.org/0000-0001-6743-803X
Heba Alghoul http://orcid.org/0000-0001-8234-5843
Carlos Areia http://orcid.org/0000-0002-4668-7069
Albert Prats-Uribe http://orcid.org/0000-0003-1202-9153
Peter R Rijnbeek http://orcid.org/0000-0003-0621-1979
Kristin Kostka http://orcid.org/0000-0003-2595-8736
Talita Duarte-Salles http://orcid.org/0000-0002-8274-0357

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
