## [Reviewer comments · BMJ Open]

ARTICLE DETAILS

TITLE (PROVISIONAL)	Characteristics and outcomes of COVID-19 patients with and without prevalent hypertension: a multinational cohort study
AUTHORS	Reyes, Carlen; Pistillo, Andrea; Fernández-Bertolín, Sergio; Recalde, Martina; Roel, Elena; Puente, Diana; Sena, Anthony; Blacketer, Clair; Lai, Lana; Alshammari, Thamir; Ahmed, Waheed-UI-Rahman; Alser, Osaid; Alghoul, Heba; Areia, Carlos; Dawoud, Dalia; Prats-Urbe, Albert; Valveny, Neus; de Maeztu, Gabriel; Sorlí Redó, Luisa; Martínez Roldan, Jordi; Lopez Montesinos, Inmaculada; Schilling, Lisa; Golozar, Asieh; Reich, Christian; Posada, Jose; Shah, Nigam; You, Seng; Lynch, Kristine; DuVall, Scott; Matheny, Michael; Nyberg, Fredrik; Ostropolets, Anna; Hripcsak, George; Rijnbeek, P; Suchard, MA; Ryan, Patrick; Kostka, Kristin; Duarte-Salles, Talita

VERSION 1 – REVIEW

REVIEWER	Armen Yuri Gasparyan Dudley Group NHS Foundation Trust (Teaching Trust of the University of Birmingham), Departments of Rheumatology and Research & Development
REVIEW RETURNED	10-Oct-2021

GENERAL COMMENTS	This is the largest multinational COVID-19 cohort study that describes some associations of arterial hypertension with COVID-19 outcomes, particularly hospitalizations and death. First of all, the authors should clarify whether ethics full review waiver was obtained from each participating centre. Additionally, approval of local health administrators/managers is required to allow processing and publicizing the obtained data. Protocol numbers and dates of approval/waiver by each participating centre are requested. Methods. How arterial hypertension was defined and diagnosed at each participating centre? The abstract needs to be revised to clearly reflect data in connection with the analyzed groups. Statistical analyses. Please clarify whether percentages (prevalence) of arterial hypertension in COVID-19 and COVID-19 plus hypertension groups were statistically compared. It would be appropriate to provide details about the outcomes related to thrombotic events leading to death (myocardial infarction and ischaemic stroke). Presumably, arterial hypertension confounded thrombotic events rather than other causes of death in COVID-19 patients. Having cardiac arrhythmias and cardiovascular events in Table 2 without myocardial infarction and Ischaemic stroke columns looks odd.
---

REVIEWER	Ana Teresa Timoteo Ctr Hosp Lisboa Cent
REVIEW RETURNED	12-Oct-2021

GENERAL COMMENTS	The present manuscript describes the results from a multinational network designed to characterize patient with and without hypertension and to assess the adverse outcomes. This is a retrospective study from 15 healthcare databases from the US, Europe and Asia. All patients diagnosed/hospitalized with COVID-19 were included and stratified by hypertension status. Follow-up was from COVID-19 diagnosis/hospitalization to death, end of the study period, or 30-days outcomes (hospitalization, adverse events or death). They included 2,851,035 diagnosed and 563,708 hospitalized patients with COVID-19. Hypertensive patients diagnosed with COVID-19 were predominantly >50-year-old and female and were frequently diagnosed with obesity, heart disease, dyslipidaemia, and diabetes. Patients with hypertension had more hospitalizations and mortality. Hospitalized patients with hypertension were more likely to have acute respiratory distress syndrome, arrhythmia, and increased mortality. Strengths and limitations were properly identified in the manuscript. In particularly, the main strength is that it encompasses the whole spectrum of COVID-19 patients and not just hospitalized patients, being less biased compared to previous studies and meta-analysis. Table 1 should include aggregated information. Although it is described in the main text, visually it could have more impact. Of particular relevance is the great variability in outcomes according to the cohort, and this deserves a more detailed comment by the authors. For instance, hospitalization in hypertensive patients is described to be in the range of 1.3 to 41.1% and death of 0.3 to 18.5%. This range is very large. The study is interesting, with a large sample size and results are solid.
--

REVIEWER	Thereza Moreira State University of Ceara, ENFERMAGEM
REVIEW RETURNED	14-Oct-2021

GENERAL COMMENTS	The article is relevant and has a robust sample that tries to elucidate hypertension as a predictor of severity in Covid-19. The article has limitations because it deals with different studies, from countries with different health systems, but the authors are aware of these limitations and assume them as a possibility. The same happens when they assume that patients treated on an outpatient basis may have been hospitalized. Despite the limitations assumed, the grandeur of the study and the way the method was conducted in order to reduce such limitations, mean that the article does have innovative content and is endowed with scientific and social relevance. For these reasons, I recommend PUBLISHING the article after improving the visibility of your figures.
---

VERSION 1 – AUTHOR RESPONSE

Reviewer: 1
Dr. Armen Yuri Gasparyan, Dudley Group NHS Foundation Trust (Teaching Trust of the University of Birmingham)

Comments to the Author:

This is the largest multinational COVID-19 cohort study that describes some associations of arterial hypertension with COVID-19 outcomes, particularly hospitalizations and death.

1- First of all, the authors should clarify whether ethics full review waiver was obtained from each participating centre.

Authors response:

We agree and understand the concerns of the reviewer regarding the ethics approvals and the sharing of data, given the multinational-multidata base nature of this study. Please see answer to the Editor (point 1 above)

This study is encompassed in the “Characterizing Health Associated Risks, and Your Baseline Disease In SARS-COV-2 (CHARYBDIS): protocol for an OHDSI network study” which protocol can be accessed through this link:

https://github.com/ohdsi-studies/Covid19CharacterizationCharybdis/blob/master/documents/Protocol_COVID-19%20Charybdis%20Characterisation_V5.docx

Authors' Action:

No further action taken (please see the included changes in the response to editors above)

2- Additionally, approval of local health administrators/managers is required to allow processing and publicizing the obtained data. Protocol numbers and dates of approval/waiver by each participating centre are requested.

Authors response:

Approval from all participating databases (or exemption of this in some cases as mentioned above) has been included in the manuscript together with the code of approval from each committee. Additionally, at least one researcher from each database participated in this manuscript as a co-author, including the review of the presented results and the acceptance of the current version of the manuscript.

Authors' Action:

No further action taken

3-Methods. How arterial hypertension was defined and diagnosed at each participating centre?

Authors' response:

All outcomes were ascertained based on the Systematized Nomenclature of Medicine Current Terminology (SNOMED CT) hierarchy used in the Common Data Model of the OMOP studies. This allowed us to use the same definition for all the databases participating in the study. The link to explore the definition used for the definition of prevalent hypertension can be accessed here: (ATLAS tool: <https://atlas.ohdsi.org/#/cohortdefinition/227>) and in the supplementary table 3. (Stated in page 11, lines 246-247). All included codes by database were also made publicly available at <https://data.ohdsi.org/Covid19CharacterizationCharybdisDiagStrata/>

However, we agree that this was not clearly stated in the methods section and for this reason we have added it both in the methods as detailed below.

Finally, it is important to mention that the diagnosis of hypertension is registered by a medical doctor in each patient's electronic health records.

Authors' Action:

Changes can be seen in page 10 lines 252-253 as follows:

"...index date and identified comorbidities in the year before the index date. Hypertension diagnosis and comorbidities (asthma, cancer, chronic kidney and liver disease, chronic ..."

4-The abstract needs to be revised to clearly reflect data in connection with the analyzed groups.

Authors' Response:

We agree with the reviewer that the abstract needs more details on the data gathered for each cohort so that it clearly reflects our methods and results.

For this reason, changes have been made accordingly in the abstract.

Authors' Actions:

Please see changes in the abstract page 4-5 (changes are coloured in red)

Design and setting: *Retrospective cohort study using 15 healthcare databases (primary and secondary electronic health care records, insurance and national claims data) from the US, Europe and South Korea, standardized to the Observation Medical Outcomes Partnership common data model. Data was gathered from 1st March to 31st October 2020.*

Participants: *Two non-mutually exclusive cohorts were defined: 1) individuals diagnosed with COVID-19 (diagnosed cohort) and 2) individuals hospitalized with COVID-19 (hospitalized cohort) and stratified by hypertension status. Follow-up was from COVID-19 diagnosis/hospitalization to death, end of the study period, or 30-days.*

Outcomes: *Demographics, comorbidities, and 30-day outcomes (hospitalization and death for the diagnosed cohort and adverse events and death for the hospitalized cohort) were reported.*

Results: *We identified 2,851,035 diagnosed and 563,708 hospitalized patients with COVID-19. Hypertension was more prevalent in the latter (range (% , 95%CI) across databases 17.4 (17.2-17.6)-61.4 (61.0-61.8) and 25.6 (24.6-26.6)-85.9 (85.2-86.6). Patients in both cohorts with hypertension were predominantly >50-year-old and female. Patients with hypertension were frequently diagnosed with obesity, heart disease, dyslipidaemia, and diabetes. Compared to patients without hypertension, patients with hypertension, in the COVID-19 diagnosed cohort, had more hospitalizations (range 1.3 (0.4-2.2)- 41.1 (39.5-42.7) vs 1.4 (0.9-1.9)-15.9 (14.9-16.9)) and mortality (0.3(0.1-0.5)-18.5 (15.7-21.3) vs 0.2 (0.2-0.2)-11.8 (10.8-12.8)). Patients in the COVID-19 hospitalized cohort with hypertension were more likely to have acute respiratory distress syndrome (0.1(0.0-0.2) -65.6 (62.5-68.7) vs 0.1 (0.0-0.2)-54.7 (50.5-58.9)), arrhythmia (0.5 (0.3-0.7)-45.8 (42.6-49.0) vs 0.4 (0.3-0.5)-36.8 (32.7-40.9)) and increased mortality (1.8 (0.4-3.2)-25.1 (23.0-27.2) vs 0.7 (0.5-0.9)-10.9 (10.4-11.4)) than patients without hypertension.*

5-Statistical analyses. Please clarify whether percentages (prevalence) of arterial hypertension in COVID-19 and COVID-19 plus hypertension groups were statistically compared.

Authors' Response:

The prevalence of hypertension was presented with their 95% CI (please see supporting table 1 in the supplementary file). Although we agree with the reviewer that it would be interesting to analyse the association between hypertension and diagnosis or hospitalization of COVID this was out of the scope of this study as it was intended to be descriptive in nature.

This was stated in page 12, lines 285-287:

“This is a descriptive study and no causal inference is intended. Multivariable regression or adjustment for confounding was therefore considered out of remit, and not included in our study.”

Authors' Action:

No further action taken

6-It would be appropriate to provide details about the outcomes related to thrombotic events leading to death (myocardial infarction and ischaemic stroke). Presumably, arterial hypertension confounded thrombotic events rather than other causes of death in COVID-19 patients.

Authors' Response:

Yes, indeed thrombotic events might be a confounder when analyzing the association between hypertension and death. However, as answered in point n°5, this study is descriptive and no causal inference were analysed. Unfortunately, not all the databases register the cause of death and therefore this information cannot be retrieved even at a descriptive level.

Authors Action:

No further action taken

7-Having cardiac arrhythmias and cardiovascular events in Table 2 without myocardial infarction and Ischaemic stroke columns looks odd.

Authors Response:

Yes, we agree with the reviewer that this needs to be clarified. We gathered a longer list of outcomes but not all of them were available for all the databases so after several revisions we decided to include only those that were available for most databases. The complete list of outcomes is publicly available at <https://data.ohdsi.org/Covid19CharacterizationCharybdis/>

The total cardiovascular event is a compound outcome which included: ischemic stroke, haemorrhagic stroke, heart failure (heart failure during hospitalization for the hospitalized cohort), acute myocardial infarction or sudden cardiac death. This is detailed in the methods section (page 11, lines 268-270), however we agree that this needs to be clarified in the notes of table 2 too.

Authors Action:

We have added the list of events included in the outcome “Total CV event” in the footnotes in table 2 page 18 as follows:

“ hypertension; †: Venous thromboembolic (pulmonary embolism and deep vein thrombosis) events; ‡: Acute respiratory distress syndrome; §: cardiovascular disease events (ischemic stroke, haemorrhagic stroke, heart failure (heart failure during hospitalization for the hospitalized cohort), acute myocardial infarction or sudden cardiac death)”*

Reviewer: 2

Dr. Ana Teresa Timoteo, Ctr Hosp Lisboa Cent

Comments to the Author:

The present manuscript describes the results from a multinational network designed to characterize patient with and without hypertension and to assess the adverse outcomes. This is a retrospective study from 15 healthcare databases from the US, Europe and Asia. All patients diagnosed/hospitalized with COVID-19 were included and stratified by hypertension status. Follow-up was from COVID-19 diagnosis/hospitalization to death, end of the study period, or 30-days outcomes (hospitalization, adverse events or death). They included 2,851,035 diagnosed and 563,708 hospitalized patients with COVID-19. Hypertensive patients diagnosed with COVID-19 were predominantly >50-year-old and female and were frequently diagnosed with obesity, heart disease, dyslipidaemia, and diabetes. Patients with hypertension had more hospitalizations and mortality. Hospitalized patients with hypertension were more likely to have acute respiratory distress syndrome, arrhythmia, and increased mortality.

Strengths and limitations were properly identified in the manuscript. In particular, the main strength is that it encompasses the whole spectrum of COVID-19 patients and not just hospitalized patients, being less biased compared to previous studies and meta-analysis.

1- Table 1 should include aggregated information. Although it is described in the main text, visually it could have more impact.

Authors' Response:

Indeed, one of the main challenges of this manuscript is to combine the different results found in each database and generate an overall result. For this reason, the results were aggregated in the main text to send a more consistent message regarding hypertension. However, we cannot obviate that there is a great variability between the databases and this also needs to be reflected in the tables and for this reason the results were shown for each database and not aggregated.

Authors' Action:

No further action taken.

2- Of particular relevance is the great variability in outcomes according to the cohort, and this deserves a more detailed comment by the authors.

For instance, hospitalization in hypertensive patients is described to be in the range of 1.3 to 41.1% and death of 0.3 to 18.5%. This range is very large.

The study is interesting, with a large sample size and results are solid.

Authors' Response:

We agree with the reviewer that the great variability in the outcomes needs to be clearly explained. However, we can only speculate the possible reasons as no further analysis besides the descriptive results reported can be carried out with this data.

We believe that this variability might be largely due to the different nature of the databases included. Those databases with mainly primary care records (such as CPRD) would have relied on the diagnosis made by primary care solely and could underestimate some of the outcomes such as hospitalizations or death occurring in a hospital setting, if not automatically reported to the primary care physicians, whereas, those databases with hospital information (such as SIDIAP) would likely have a lower proportion of underreporting of these events. We agree that this needs to be further acknowledged in the limitation section and for this reason, changes have been made accordingly.

Another reason for the large variability can rely in the different protocols applied for the diagnose of COVID-19 in the different countries; patients with a greater burden of comorbidities might have been directly referred to hospital settings for the diagnosis of COVID-19 bypassing the primary care centres. This would lead to a lower proportion of patients in the databases who only nourishes from primary care data. Again, the proportion of these patients cannot be retrieved from our data and therefore remains unknown.

This has been detailed in the limitation section page 21 lines 406-408: “...*minimize the impact of incompleteness. Differential reporting in databases is likely due to different coding practices as well as variability in disease severity, with milder/less symptomatic cases more likely being only diagnosed, and more severe ones hospitalized.*”

Authors' Action:

Changes can be seen in page 21 line 402

“...*different coding practices, different primary and secondary level data availability, as well as...*”

Reviewer: 3

Dr. Thereza Moreira, State University of Ceara

Comments to the Author:

The article is relevant and has a robust sample that tries to elucidate hypertension as a predictor of severity in Covid-19. The article has limitations because it deals with different studies, from countries with different health systems, but the authors are aware of these limitations and assume them as a possibility. The same happens when they assume that patients treated on an outpatient basis may have been hospitalized. Despite the limitations assumed, the grandeur of the study and the way the method was conducted in order to reduce such limitations, mean that the article does have innovative content and is endowed with scientific and social relevance.

- 1- For these reasons, I recommend **PUBLISHING** the article after improving the visibility of your figures.

Authors' Response:

We appreciate the reviewers' comments and the interest for this study. We have improved the visibility of the figures as suggested by the reviewer.

Author' Action:

Please see changes in figure 1, 2, 3 and 4.

VERSION 2 – REVIEW

REVIEWER	Ana Teresa Timoteo Ctr Hosp Lisboa Cent
REVIEW RETURNED	08-Nov-2021
GENERAL COMMENTS	My concerns were properly adressed by the authors and in my opinion, the suggestions from other reviewers were also managed properly